# Telomere-to-Telomere Haplotype-Resolved Genomes of *Agrocybe chaxingu* Reveals Unique Genetic Features and Developmental Insights

**DOI:** 10.3390/jof10090602

**Published:** 2024-08-25

**Authors:** Xutao Chen, Yunhui Wei, Guoliang Meng, Miao Wang, Xinhong Peng, Jiancheng Dai, Caihong Dong, Guanghua Huo

**Affiliations:** 1Jiangxi Key Laboratory for Excavation and Utilization of Agricultural Microorganisms, Jiangxi Agricultural University, Nanchang 330045, China; 18146618712@163.com; 2State Key Laboratory of Mycology, Institute of Microbiology, Chinese Academy of Sciences, Beijing 100101, China; menggl@im.ac.cn (G.M.); mwang2136@gmail.com (M.W.); 3Jiangxi Provincial Key Laboratory of Agricultural Non-Point Source Pollution Control and Waste Comprehensive Utilization, Institute of Agricultural Applied Microbiology, Jiangxi Academy of Agricultural Sciences, Nanchang 330200, China; yunhuiwei@sina.com (Y.W.); m15270803147@163.com (X.P.); 18770170800@163.com (J.D.); 4Key Laboratory of Crop Physiology, Ecology and Genetic Breeding Ministry of Education, Jiangxi Agricultural University, Nanchang 330045, China

**Keywords:** *Agrocybe chaxingu*, sexually compatible monokaryons, telomere-to-telomere genomes, mating-type, unique genes

## Abstract

*Agrocybe chaxingu* is a widely cultivated edible fungus in China, which is rich in nutrients and medicinal compounds. However, the lack of a high-quality genome hinders further research. In this study, we assembled the telomere-to-telomere genomes of two sexually compatible monokaryons (CchA and CchB) derived from a primarily cultivated strain AS-5. The genomes of CchA and CchB were 50.60 Mb and 51.66 Mb with contig N50 values of 3.95 Mb and 3.97 Mb, respectively. Each contained 13 complete chromosomes with telomeres at both ends. The high mapping rate, uniform genome coverage, high LAI score, all BUSCOs with 98.5%, and all base accuracy exceeding 99.999% indicated the high level of integrity and quality of these two assembled genomes. Comparison of the two genomes revealed that approximately 30% of the nucleotide sequences between homologous chromosomes were non-syntenic, including 19 translocations, 36 inversions, and 15 duplications. An additional gene *CchA_000467* was identified at the *Mat A* locus of CchA, which was observed exclusively in the *Cyclocybe cylindracea* species complex. A total of 613 (4.26%) and 483 (3.4%) unique genes were identified in CchA and CchB, respectively, with over 80% of these being hypothetical proteins. Transcriptomic analysis revealed that the expression levels of unique genes in CchB were significantly higher than those in CchA, and both CchA and CchB had unique genes specifically expressed at stages of mycelium and fruiting body. It was indicated that the growth and development of the *A. chaxingu* strain AS-5 required the coordinated action of two different nuclei, with CchB potentially playing a more significant role. These findings contributed to a more profound comprehension of the growth and developmental processes of basidiomycetes.

## 1. Introduction

*Agrocybe chaxingu* is a widely cultivated edible mushroom in Asia that belongs to the Strophariaceae family. It is a member of the *Cyclocybe cylindracea* species complex, which includes *Cyclocybe cylindracea*, *Cyclocybe aegerita*, and *Cyclocybe salicaceicola* [1]. In 2022, the production of *A. chaxingu* in China reached 882,300 tonnes, ranking 8th among all the cultivated mushrooms, as reported by the China Edible Fungi Association (https://www.cefa.org.cn/web/index.html, accessed on 28 December 2023).

Consumers have a heightened appreciation for the *A. chaxingu* due to its unique flavor and nutritional value [2]. In addition, *A. chaxingu* was reported to have various pharmacological effects, such as anti-inflammatory [3], hypoglycemic [4], and anti-tumor activities [5]. Furthermore, the ribotoxin “Ageritin” was discovered in *C. aegerita* [6] and reported to exhibit ribonucleolytic activity on ribosomes, ribonuclease activity on tobacco mosaic virus RNA, endonuclease activity on plasmid and genomic DNAs, as well as antiproliferative and defense properties [7]. In 2004, the fungal heme-thiolate enzyme subfamily of unspecific peroxygenases were first described in *C. aegerita* [8], which were versatile biocatalysts oxidizing non-activated C-H bonds in a regiospecific and stereospecific manner under mild reaction conditions [9]. The plethora of bioactive compounds produced by the *C. cylindracea* species complex requires further comprehensive research. The availability of high-quality genomes will facilitate the exploration of secondary metabolism.

In addition, several species within the *C. cylindracea* species complex are morphologically similar. The phylogenetic relationships among *A. chaxingu*, *C. cylindracea*, *C. aegerita*, and *C. salicaceicola* were complex [10]. Phylogenomic analysis can facilitate the elucidation of the phylogenetic relationships, evolutionary histories, and population structures of these species. It is crucial to clarify the phylogenetic relationships between *A. chaxingu* and other taxa within the *C. cylindracea* species complex for the future breeding of *A. chaxingu*. The genome of the *C. aegerita* strain AAE-3 was published in 2018, unveiling a repertoire of fruiting-related genes and biopolymer-degrading enzymes [11]. Subsequently, the genome of *C. cylindracea* was assembled, and detailed pathways for the synthesis of nutritional and flavor-related compounds were analyzed [12]. These genomes within the *C. cylindracea* species complex comprised of 122 and 3790 scaffolds. In 2023, a haploid genome draft of the *A. chaxingu* strain MP-N11 was reported to be assembled (Submitted GenBank assembly: GCA_027627235.1, https://www.ncbi.nlm.nih.gov/datasets/genome/GCA_027627235.1/, accessed on 22 May 2023). However, the assembly consisted of 4634 contigs with an N50 of 18.1 Kb, indicating that there is room for improvement in the assembly quality.

*A. chaxingu* is a tetrapolar heterothallic basidiomycete fungus [13]. During most of its lifecycle, each cell contains two nuclei with different mating types, and these two nuclei share the same cytoplasm. The current incomplete haploid genome of *A. chaxingu* obviously cannot fully represent the entire genetic information of the species. In addition, the chromosomal structural variances between these two nuclei and their interactions are critical for understanding the growth and development mechanisms of mushroom-forming fungi.

In this study, whole-genome sequencing of two sexually compatible monokaryons (CchA and CchB) derived from the dominant cultivator *A. chaxingu* strain AS-5 was performed. The telomere-to-telomere (T2T) chromosome-level genomes of CchA and CchB were generated by incorporating data from the Illumina, PacBio high fidelity (HiFi), and High-throughput Chromatin Conformation Capture (Hi-C) sequencing platforms. Subsequently, comparative genomic analysis revealed differences between the reference genomes of the two monokaryotic strains with compatible mating types in terms of chromosomal structural variations, mating loci, and gene family evolution. Finally, an analysis was conducted on the gene expression profiles at the mycelial and fruiting body stages of the *A. chaxingu* AS-5, with a particular focus on the unique genes of the strains CchA and CchB. This study contributed to understanding the genetic differences between the two nuclei of *A. chaxingu* and their roles in fruiting body development through T2T genome comparisons, mating type loci analysis, and transcriptional profiling at two stages.

## 2. Materials and Methods

### 2.1. A. chaxingu Strain, Culture Conditions, and Nucleic Acid Preparation

The strain AS-5 is the commercially dominant cultivar of *A. chaxingu* in China. It was preserved at the China General Microbiological Culture Collection Center (CGMCC No. 40980). The monokaryotic strains of the dikaryotic strain AS-5 were isolated through protoplast monokaryotization following our previously reported method [14]. The detailed procedure was as follows: Seven fungal plugs (4 mm in diameter) of *A. chaxingu* AS-5 were inoculated onto a Potato Dextrose Agar (PDA) plate (9 cm in diameter) that had been covered with cellophane. The plate was incubated in the dark at 25 °C until the fungal mycelia of the colonies came into contact with one another. Next, the cellophane with the fungal colonies was transferred to a new sterile Petri dish using sterile forceps, ensuring the mycelium side faced upwards. A 3 mL solution of 1.5% (*w*/*v*) lywallzyme (Guangdong Microbiology Institute) was then applied to the mycelia. The solution was prepared using a 0.6 mol/L mannitol solution. Enzymatic digestion was conducted at 33 °C for 70 min to obtain protoplasts, which were subsequently counted using a hemocytometer. Then, 50 μL of the enzyme solution containing 10^5^ protoplasts/mL was spread onto regeneration medium plates, which were prepared with 200.00 g of potato, 20.00 g of glucose, 20.00 g of agar, 205.38 g of sucrose, and 1.00 g of maltose, made up to a final volume of 1000 mL. The plates were incubated in the dark at 25 °C. After the mycelia had developed, single colonies were picked and transferred to PDA plates. The monokaryotic strains without any clamp connection were selected under 1000× magnification using an optical microscope (Eclipse 80i, Nikon, Tokyo, Japan). Strains CchA and CchB were two monokaryotic strains with compatible mating types. They were capable of forming a clamp connection when co-cultured. Both the dikaryotic and monokaryotic strains were cultivated in the dark on PDA medium at 25 °C.

For the genome sequencing, the monokaryotic strains CchA and CchB were cultured in Potato Dextrose Broth (PDB) at 25 °C in the dark for 14 d. For the transcriptome sequencing, the mycelia of *A. chaxingu* AS-5 were cultured on a PDA medium for 7 d. Under conditions of 25 °C and scattered light, *A. chaxingu* AS-5 formed primordia on the substrate (63% sawdust, 20% cottonseed hulls, 15% bran, 1% lime, and 1% gypsum) and then grew for an additional 5 d to form mature fruiting bodies. Then, the mycelia and fruiting bodies were collected and washed with sterile water, and promptly frozen in liquid nitrogen for subsequent nucleic acid extraction. Genomic DNA was extracted using the QIAGEN^®^ Genomic Kit (Qiagen, Dusseldorf, Germany) according to the manufacturer’s instructions. Total RNA extraction was performed using TRIzol reagent (Invitrogen, Carlsbad, CA, USA).

### 2.2. Genome Sequencing and Heterozygosity Estimation

Both the DNA samples from strains CchA and CchB were sequenced using the MGISEQ-T7 (with the construction of a 350 bp library) and the PacBio Sequel II (with the PacBio HiFi construction of a 15 Kb SMRTbell library) platforms at Wuhan Grandomics Biosciences Co., Ltd. (Wuhan, Hubei, China). The DNA of strain CchA was subjected to Hi-C sequencing on the Illumina NovaSeq 6000 platform at Shanghai Biozeron Biotech. Co., Ltd. (Shanghai, China), with paired-end reads of 150 bp. The RNA from the mycelia of the strains CchA and *A. chaxingu* AS-5 and the fruiting bodies of the strain *A. chaxingu* AS-5 were used for transcriptome sequencing on the Illumina NovaSeq 6000 platform at Shanghai Biozeron Biotech. Co., Ltd. (Shanghai, China).

For the estimation of genome heterozygosity, Jellyfish v 2.1.3 [15] was utilized for K-mer distribution analysis based on next-generation sequencing reads (NGS). The results were then imported into GenomeScope v1.0 [16] with a K-mer length of 21 and a ploidy of 2.

### 2.3. De Novo Genome Assembly and Quality Assessment

The de novo genome assembly of the strains CchA and CchB primarily involves error correction to identify haplotypes, the construction of assembly graphs, and the generation of assembled sequences. Firstly, we used Smrt link v7.0 (https://github.com/WenchaoLin/SMRT-Link, accessed on 6 July 2023) to pre-process the PacBio raw reads, removing adapter sequences, low-quality regions, and sequences containing sequencing errors. We also employed the default parameters of the CCS software v3.0.0 (https://github.com/PacificBiosciences/ccs, accessed on 10 July 2023) to generate HiFi reads. Using the Hifiasm v0.13 software [17] with default parameters, we performed de novo assembly of PacBio HiFi long-reads. Subsequently, short reads were employed and further error correction of the assembled genome was conducted using the Pilon v1.23 [18]. ALLHIC software (https://github.com/tanghaibao/allhic, accessed on 21 August 2023) was used to link the genome contigs or scaffolds.

To address the missing telomeres, a manual identification process was used to extract the telomeric repeat sequences ‘CCCTAA’ and ‘TTAGGG’ from the PacBio HiFi reads. The extracted sequences of the missing telomeres were assembled using the Hifiasm software v0.12 [17] with default parameters, and they were subsequently matched with each chromosome. Further error correction was performed using Racon [19] and Pilon [18] to ensure assembly quality.

The quality assessment of each genome primarily included the long terminal repeat (LTR) assembly index (LAI) [20], mapping rate of reads, overall base accuracy estimates, and Benchmarking Universal Single-Copy Orthologs (BUSCO) completeness analysis [21].

### 2.4. Genome Annotation

De novo prediction, homology-based prediction, and transcriptome prediction were employed for gene prediction. De novo gene prediction was performed using Augustus v3.3.1 [22]. For homology prediction, the protein sequences of *Psilocybe cubensis* (the National Center for Biotechnology Information GenBank: GCA_017499595.2), *Agrocybe pediades* (GCA_015484485.1), *Pholiota molesta* (GCA_014925825.1), and *Hypholoma sublateritium* (GCA_000827495.1) were mapped to the *A. chaxingu* genome using GeMoMa v1.6.2 [23] with default parameters. To facilitate gene prediction using transcriptome sequencing (RNA-seq), we used PASA v2.3.3 [24] to map RNA-seq data onto the *A. chaxingu* genome. Finally, EVM [25] was used to integrate these data to eliminate redundancy in the final gene set.

We used the rRNAmmer v1.2 [26] to detect rRNA in the *A. chaxingu* genome by ab initio prediction and homologous sequences downloaded from the Ensembl database [27]. The tRNAscan-SE with the default parameters was employed to identify the genes related to tRNA [28]. For non-coding RNA such as snRNA and miRNA, we utilized the Infernal [29] with default parameters for annotation, based on the Rfam database [30].

RepeatModeler v2.0.2a [31] and RepeatMasker v4.1.2 [32] softwares were utilized to predict transposon sequences by aligning the assembled genome of *A. chaxingu* against the transposon Repbase database [33]. Additionally, LTR_ finder [34] and LTR_ harvest [35] were employed for de novo searching of LTR retrotransposons. The unknown repetitive sequences were classified using DeepTE [36].

For functional annotation of the predicted protein-coding genes, BLASTP v2.7.1 [37] was used to align protein sequences against databases including Protein FAMilies Database (PFAM) [38], Integrated Resource of Protein Domains and Functional Sites (InterPro) [39], Clusters of Orthologous Groups (COGs) [40], Gene Ontology Terms (GO Terms) [41], BUSCO [21], and Evolutionary Genealogy of Genes: Non-supervised Orthologous Groups (EggNOG) [42]. InterProScan 5.32-71.0 [43] was used to identify protein domains and obtain the annotations of GO using default parameters. Kyoto Encyclopedia of Genes and Genomes (KEGG) annotation was performed using KAAS [44].

The carbohydrate-active enzymes (CAZymes) were analyzed using the CAZyme database (http://www.cazy.org/, accessed on 25 August 2023). Fungal AntiSMASH 7.1.0 (https://fungismash.secondarymetabolites.org/, accessed on 2 September 2023) was utilized to predict secondary metabolite gene clusters.

### 2.5. Identification of Mating-Type Locus

The identification of mating-type A (*MatA*) locus involved utilizing genome annotation results, conducting homology searches in closely related species, and performing homology searches in model species for mitochondrial intermediate peptidase (Mip)-associated genes related to the *MatA*. The selected closely related species included *C. cylindracea* [12] and *C. aegerita* [11], while the model species chosen were *Schizophyllum commune* [45] and *Coprinopsis cinerea* [46]. The pheromone receptor (*Pr*) genes of the mating-type B locus (*MatB*) were searched for in the annotation results of GO Terms. Simultaneously, the two highly conserved *Pr* genes reported in the closely related species *C. salicacola* [47] were used for a homology search to determine the location of *MatB*, while other Pr proteins were validated through a BLASTP homology comparison in the *C. aegerita* genome [11]. The sequence length of pheromone precursor (Pp) is typically less than 60 amino acids, and their high variability makes them difficult to identify by BLAST homology searches. Furthermore, gene prediction methods can only predict a small fraction of these genes. Based on the localization of *Pp* genes in proximity to the *Pr* [48,49,50], the presence of the CAAX-box motif at the 3′ end, and upstream conserved AF and EA motifs [51,52,53,54], we annotated the *Pp* genes by utilizing Expasy (https://web.expasy.org/translate/, accessed on 15 September 2023) to annotate the DNA sequences within a 50 kb range of the *Pr*.

To verify the authenticity of the gene *CchA_000467* at *MatA* locus in the strain CchA, primers (CchA-F: GCTTGGGTGACGTCATTAC/CchA-R3: CARATTGAWGTTTGGGTAC) were designed. PCR reaction was performed in 20 µL of total volume: 0.8 µL of each primer (10 µM), 2 µL of DNA, 10 µL of PrimeSTAR^®^ Max DNA Polymerase (TAKARA), and 6.4 µL of ddH2O. The PCR was performed as follows: 95 °C, 3 min; 35 cycles of 15 s at 95 °C, 15 s at 48 °C, and 55 s at 72 °C; 72 °C, 10 min. The PCR products were checked through electrophoresis on 1.0% agarose gel.

### 2.6. Phylogenomic and Evolutionary Analyses

Phylogenomic analysis was performed with 15 fungal genomes (Appendix A). We identified single-copy orthologous genes using OrthoFinder v2.5.4 [55] and performed sequence alignment using Muscle v3.8.31 [56] with default parameters. Conservative sequences were selected using Gblocks v0.91b [57] and subsequently concatenated into a supermatrix by seqkit v2.2.0 [58]. A maximum-likelihood phylogenetic tree was constructed using RAxML-NG v0.9.0 [59] with the amino acid substitution matrix LG+F+I+G4 selected by ProtTest v3.4.2 [60]. The optimal tree was visualized in FigTree v1.4.4 (https://github.com/rambaut/figtree/, accessed on 6 January 2024). The software r8s v1.81 [61] was utilized to estimate the divergence times of species with an approximate likelihood method by calibrating with two fossil record fungal species of *C. cinerea* and *Laccaria bicolor* [62]. CAFE5 v1.1 [63] was employed to calculate expansions or contractions in gene families.

### 2.7. Comparative Genomic Analysis

Whole-genome synteny analysis of the nucleotide sequences was executed on the monoploid genomes of strains CchA and CchB using MUMmer [64]. The gene collinearity relationships between each chromosome of CchA and CchB were analyzed using MCScanX (E value < 1 × 10^−5^) [65]. Furthermore, we conducted an analysis using SyRI v1.5 software [66] to identify structural variations larger than 10 kb between homologous chromosomes of two genomes, including translocations, inversions, and duplications. The identification of unique genes between the genome of strains CchA and CchB was carried out using OrthoFinder v2.5.4 [55]. Subsequently, the KEGG enrichment of these specific genes was analyzed using TBtools [67].

To verify the authenticity of the unique genes *CchA_005693* and *CchA_005699* in the genome of CchA, primers (005693-F: ATGCTGACTTTGAGATATAAGAAGCAG/005693-R: TTATTTCGCATTGGGTGGCG; 005699-F: ATGAAAGGTGGGATGTCGAG/005693-R: TTATTTCGCATTGGGTGGCG, respectively) were designed. PCR reaction was performed in 20 µL of total volume: 0.8 µL of each primer (10 µM), 2 µL of DNA, 10 µL of PrimeSTAR^®^ Max DNA Polymerase (TAKARA), and 6.4 µL of ddH2O. The PCR was performed as follows: 95 °C, 3 min; 35 cycles of 15 s at 95 °C, 15 s at 55 °C, and 15 s at 72 °C; 72 °C, 10 min. The PCR products were checked through electrophoresis on 1.0% agarose gel.

### 2.8. Transcriptome Analysis

The relative expression levels were recorded as transcripts per kilobase of exon model per million mapped reads (TPM), which were calculated using featureCounts 2.0.6 [68]. The clean RNA-seq reads were mapped to the assembly genomes of strains CchA and CchB using HISAT v2.2.1 [69] for TPM calculation.

## 3. Results

### 3.1. Genome Sequencing, T2T Assembly, and Quality Assessment

Strains CchA and CchB were two monokaryotic strains with compatible mating types isolated through protoplast monokaryotization from the commercially dominant cultivar “AS-5” (Figure 1A–H). They were capable of forming a clamp connection when co-cultured. Genome sequencing of the monokaryotic strains CchA and CchB (Appendix A) was conducted using the MGISEQ-T7 and PacBio Sequel II platforms. A total of 9.55 Gb (sequence coverage ~193.26×) and 6.99 Gb (~138.56×) of NGS reads were generated for strains CchA and CchB, along with 7.30 Gb (~147.73×) and 7.04 Gb (~139.55×) of PacBio HiFi reads, respectively. In addition, Hi-C sequencing of the strain CchA was conducted on the Illumina NovaSeq 6000 platform, generating 5.04 Gb (~101.89×) of clean data. Based on 21 K-mer analysis (Appendix A), it was found that heterozygosity levels of strains CchA and CchB were only 0.03% and 0.04%, respectively. The estimated genome sizes were 45.60 Mb for strain CchA and 47.31 Mb for strain CchB.

We successfully assembled two T2T chromosome-level haploid genomes of *A. chaxingu* (Figure 2). The CchA genome had a total length of 50.60 Mb (50,604,502 bp) with 13 chromosomes, N50 of 3.95 Mb, and chromosome lengths ranging from 2.65 Mb to 5.19 Mb (Appendix A). The CchB genome measured 51.66 Mb (51,660,018 bp), with 13 chromosomes, N50 of 3.97 Mb, and chromosome lengths ranging from 2.98 Mb to 5.46 Mb (Appendix A). The GC content of the CchA genome was 50.99%, while that of CchB was 50.98%. A total of 26 telomeres were detected at both ends of their 13 chromosomes, characterized by (CCCTAA)_n_ and (TTAGGG)_n_ repeats for the genomes of both strains.

Ribosomal DNA (rDNA) was predicted at the end of the chromosome 12 (chr12) of both the CchA and CchB genomes (Appendix A). In CchA, the closest Internal Transcribed Spacer (ITS) to the telomere was located 10 kb away, whereas in CchB, the nearest ITS was only 1 kb from the telomere sequences.

The assembly quality and completeness of the CchA and CchB genomes were assessed through various methods. Initially, consistent with the k-mer evaluation, the CchB genome surpassed that of CchA by over 1 Mb. The visualized Hi-C data for the CchA genome exhibited a high degree of consistency across all chromosomes (Appendix A), indicating the accuracy of their ordering and orientation. Secondly, aligning the HiFi reads from CchA and CchB to the two assembled genomes revealed an average HiFi read coverage of approximately 142.9× and 90× for each chromosome, respectively (Appendix A). The particularly high coverage at the end of chr12 was due to the rDNA sequences. The respective LAI scores for the genomes of CchA and CchB were 23.35 and 25.76, reaching gold quality [20]. Moreover, 99.56% and 99.96% of Illumina reads, as well as 100% and 99.92% of HiFi reads, were successfully mapped to the genomes (including the mitochondrial genome) of CchA and CchB, respectively, indicating sufficiently high coverage (Appendix A). Using Merqury v1.3 [70], the accuracy of the genome was assessed. The results showed that in the genome of CchA, out of 34,533,832 k-mers (k-mer = 18), only 2611 k-mers were found in the genome but not in the HiFi reads, with a quality value (QV) of 53.5187. The overall base accuracy estimated by T2T assembly was 99.9996%. In the genome of CchB, out of 38,006,749 k-mers (k-mer = 18), only 6606 k-mers were found in the genome but not in the HiFi reads, with a QV of 49.9034. The overall base accuracy estimated by T2T assembly was 99.9990%. Finally, BUSCO analyses revealed that both genomes exhibited a completeness of 98.5%, indicating a high level of genome integrity (Table 1). Overall, these results demonstrated the high quality, accuracy, and reliability of the reference genome for both strains CchA and CchB.

As shown in Table 2, in comparison with the other five genomes within the *C. cylindracea* species complex, the assembly quality of CchA and CchB has been significantly improved, achieving a chromosome-level genome.

### 3.2. Repeat Annotation, Noncoding RNA Annotation, and Gene Prediction

We identified 8.59 Mb (16.97%) and 9.28 Mb (17.97%) of repetitive sequences in the genome assemblies of the strains CchA and CchB, respectively (Table 3). Among the annotated and classified repetitive sequence elements, the dominant included LTR sequences and Long Interspersed Nuclear Elements (LINEs). In the genome of CchA, LTRs accounted for 3.45% (1.75 Mb), while in CchB, the proportion was 4.36% (2.25 Mb). LTRs comprised two subtypes, *Copia* and *Gypsy*. Additionally, LINEs represented 1.73% (0.87 Mb) of the CchA sequence and 1.66% (0.86 Mb) of the CchB sequence. A total of 9.28% (4.97 Mb) and 10.18% (5.26 Mb) of repetitive elements were unclassified in the genome of CchA and CchB, respectively. Furthermore, the CchA and CchB genomes exhibited distinct repetitive elements. Specifically, the L1/CIN4 and Tc1-IS630-Pogo elements were exclusive to CchA, whereas the RTE/Bov-B and Hobo-Activator elements were found solely in CchB. Notably, the Hobo-Activator repetitive element was more prevalent, comprising 0.29% (0.15 Mb) of the CchB genome. Transposons are mobile genetic elements and an important source of gene variation and evolution in many organisms [71], such as Hobo-Activator participating in chromosome rearrangement or breakage [72].

LINEs were highly concentrated in a certain position across all chromosomes of CchA and CchB, primarily spanning a range of 100 to 600 Kb (Figure 3A,B), with minimal presence in other regions. In these regions, both GC content and gene density were relatively low. The *Copia* LTRs were mainly concentrated in the same regions where LINEs were prevalent. The regions with low GC content, low gene density, and high repetitive sequence content (especially concentrated LINEs) might have represented centromeres (Figure 2 and Figure 3). The predicted centromere regions in the genome of strain CchA were confirmed in its Hi-C heatmap (Appendix A). These characteristics exhibited some similarities with the putative centromeric regions of *Tricholoma matsutake* (where LINE-rich regions were GC-rich) [73].

For non-coding RNA (Appendix A), 0.0686% (0.03 Mb) and 0.0719% (0.04 Mb) were identified in the genomes of strains CchA and CchB, respectively.

The assembled T2T genomes of strains CchA and CchB were predicted to contain 14,376 and 14,207 protein-coding genes, respectively, and 3.05% (438) and 3.00% (427) CAZyme genes were identified in each strain (Appendix A).

### 3.3. Analysis of Mating-Type Genes

By combining homology-based search and gene function annotation results, the *Mip* gene was identified. Subsequently, two reverse-oriented genes encoding homeodomain (Hd) proteins adjacent to the *Mip* gene in the CchA and CchB genomes were detected both on chr1 (Figure 4A). In *S. commune* [45] and *C. cinerea* [74,75], multiple *Hd* genes constituted the two sub-loci of *MatA*, designated *MatAα* and *MatAβ.* In species such as *Agaricus bisporus* [76], *Lentinula edodes* [77], *Pleurotus ostreatus* [78], and *Flammulina filiformis* [79,80], additional genes containing Hd domain were found located outside the *MatA* locus. There were no other Hd1 and Hd2 domains identified apart from the *MatA* locus in the two *A. chaxingu* strains. On the other side of the *Hd* gene, a highly conserved glycosyltransferase family 8 gene (*Glgen*) was identified. No *β-flanking* (*β-fg*) genes were found in the regions upstream or downstream of the *MatA* locus in the CchA and CchB genomes. However, they were identified on the chr11 of CchA and CchB genomes, indicating that *β-fg* genes are no longer directly linked with the *MatA* locus in *A. chaxingu*. A similar phenomenon has been observed in *F. filiformis* [80] and *Sparassis latifolia* [81].

A comparative synteny analysis of the *MatA* region among several major edible fungi with annotated genomes revealed that over 70% of the genes in the *MatA* region were conserved, although they exhibited some degree of variation in their relative positions (Appendix A). Furthermore, it was demonstrated that in *A. bisporus* H97, *P. ostreatus* PC9, and *P. eryngii* ATCC90797, the *MatA* region was surrounded by *Mip* genes and *β-fg* genes. In *L. edodes* L808-1, the *Mip* gene and *β-fg* gene in the *MatA* region were found to be located on the same side of the chromosome. However, in the *C. cylindracea* species complex, including *A. chaxingu* and *C. aegerita*, neither the *MatA* region nor the entire chromosome exhibited the presence of *β-fg* genes.

Through a comparative analysis of the collinearity of *MatA* in the CchA and CchB genomes, it was observed that the similarity of Hd1 proteins with the two genomes was 32.77%, and that of the Hd2 proteins was 35.87%. Additionally, there was no similarity in the intergenic regions between *Hd1* and *Hd2* genes. The similarity of most other proteins near the *MatA* locus exceeded 95% between the two genomes. On the chr11 of the CchA and CchB genomes, the similarity of β-fg proteins was 99.04%.

Compared to CchB, the *MatA* locus in the CchA genome contains an additional gene, *CchA_000467*, located between the *Hd2* and *Glgen* genes. The existence of the *CchA_000467* gene was supported by 122 reads in the original HiFi sequences and was verified through PCR amplification (Appendix A). After conducting a homology search for the *CchA_000467* gene, it was found to have similarity to a nuclear receptor coregulator, suggesting that CchA_000467 might have been involved in the process of co-activating or repressing effective transcriptional regulation by nuclear receptors [82].

Further analysis revealed that there was a *CchA_000467* homologous gene on chr9 in the genomes of both CchA (*CchA_010591*) and CchB (*CchB_011590*). Two proteins, CchA_000467 and CchA_010591, exhibited a 68.74% similarity in strain CchA. Subsequently, we investigated the presence of *CchA_000467* in other *A. chaxingu* strains. It was discovered that similar to strain CchA, haploid strain *A. chaxingu* NLJ89 harbored two proteins in its genome: one located adjacent to Hd2 and the other situated on a different chromosome. Conversely, only one homologous gene of *CchA_000467* was found in close proximity to *Hd2* in *C. aegerita* AAE3.

The homologous proteins of CchA_000467 were also found in the genomes of other basidiomycetes (Appendix A). The position of these genes was observed to include a considerable distance from the *MatA* locus, adjacent from the *MatA* locus, a separation from the *MatA* locus by multiple additional genes, and a separation from the *MatA* locus by one or two genes. The occurrence of *CchA_000467* or its homologs close to *Hd2* has thus far been observed exclusively within the *C. cylindracea* species complex.

Based on the amino acid sequences of CchA_000467 and its homologs from other species (Appendix A), a phylogenetic tree was constructed (Figure 5). The five groups were clustered with strong support as clade I and II, clade III, clade IV, clade V, and clade VI. The division of groups aligns with the classification of this gene and its positional relationship with *MatA* locus.

In previous studies, the majority of published genome sketches were of a single haploid or diploid, without any accompanying description of the other compatible haploid. In this study, the mating type structure of another haploid, CchB, which is compatible with CchA, was clearly identified, and it was demonstrated that CchB lacks the *CchA_000467* gene or its homologs in *MatA* locus.

Two *Pr* genes, *Pr1* and *Pr2*, were identified on chr10 in the CchA genome, and homologous genes were also found on chr10 in the CchB genome (Figure 4B). These two Pr proteins in the two genomes showed strong conservation with a similarity of 96.62% and 99.37%, respectively. Three other *Pr* genes, *Pr3*, *Pr4*, and *Pr5*, on chr10 of the CchA and CchB genome were identified. However, there was low similarity for these proteins (57.90% for Pr3, while other similarities ranged between 20% and 50%). In addition, a *Pr6* gene was identified on chr3 in both the CchA and CchB genomes with a protein similarity of 99.78%. Each of the six *Pr* genes in the CchA and CchB genomes has been validated in gene functional annotation. This study identified six *Pr* genes, with only *Pr6* located independently. A *Pr* gene situated outside the mating locus has also been reported in the genome of *S. commune*, where four Pr-like genes were found: *brl1*, *brl2*, and *brl3* are located at the B mating-type locus, while *brl4* is positioned separately [83].

Three *Pp* genes were successfully identified on chr10 of the CchA and CchB genome. Only Pp1 proteins showed a similarity of 88.89% between the two genomes. No similarity was observed among the other Pp proteins, indicating the differences in the *MatA* and *MatB* loci between the strains CchA and CchB.

The collinearity changes in genes within the *MatB* locus were also notably distinct, with significant variations in gene order and quantity observed among different species. In this study, the *MatB* of *A. chaxingu* was similar to, but not identical to that of *L. edodes* [77,84,85] and *F. filiformis* [79,80]. In *L. edodes*, the *MatB* locus is subdivided into two sub-loci, Bα and Bβ, each containing one *Pr* gene and one or two *Pp* genes. Additionally, there were two non-mating-type specific *Pr* genes located 12 kb away from the *MatB*, and three *Pr* genes without accompanying *Pp* genes were also found on the other scaffold [85,86]. In *F. filiformis*, there are two typical *MatB* sub-loci spaced 181 kb apart. At positions 442 kb and 138 kb from the sub-locus, two putative *Pr* genes were found.

Prior to the acquisition of the complete genome sequence, several scholars researched the genetic patterns of the *C. cylindracea* species complex through monokaryotic crosses [13,87,88]. This study revealed that the mating type genes of the *A. chaxingu* strain closely resembled those of its related species, *C. cylindracea* [12] and *C. aegerita* [89]. However, it was observed that *Hd2* in the *MatA* of *A. chaxingu* displayed an opposite orientation compared to *C. cylindracea*. Additionally, there were slight differences in the arrangement and number of *MatB* genes in comparison to both species. A profound understanding of the molecular genetic structure of mating-type systems, which govern hybridization and sexual reproduction processes, will aid in revealing the regulatory function of mating-type genes in fruiting body development. This comprehension will also help address breeding-related scientific challenges in the edible mushroom industry [90].

### 3.4. Prediction of Secondary Metabolite Biosynthetic Gene Clusters 

The secondary metabolite biosynthetic gene clusters (SM-BGCs) of the CchA and CchB genomes were predicted using antiSMASH version 7.1.0. In the genome of CchA, 33 SM-BGCs were identified, while CchB contained 34 (Appendix A). For CchA, there were 17 terpenes, five fungal ribosomally synthesized and post-translationally modified peptides (Fungal-RiPPs), five non-ribosomal peptide synthases (NRPSs), two indoles, one nonribosomal iron-siderophore (NI-siderophore), one type I polyketide synthase (T1PKS), along with two others. On the other hand, in CchB, predictions included 20 terpenes, five NRPSs, four fungal RiPPs, one indole, one NI-siderophore, one T1PKS, and two others. Notably, within CchA, an extra fungal-RiPP-like biosynthetic gene cluster was observed on chr8, and an additional indole biosynthetic gene cluster was found on chr5. Meanwhile, CchB exhibited one supplementary terpene synthase biosynthetic gene cluster on each of chr8, chr9, and chr10.

Obviously, *A. chaxingu* has significant potential for terpene biosynthesis. Terpenes are a large class of natural products with a highly diverse range of structures, widely used in pharmaceuticals, food, chemicals, and industrial raw materials [91]. In this study, the cluster of *CchA_001922*, *CchA_009732* in strain CchA, and *CchB_000247*, *CchB_009556* in strain CchB were predicted to be responsible for δ-cadinene biosynsis. δ-cadinene is an important sesquiterpene with the ability to fight against the malaria vector *Anopheles stephensi* [92], inhibit the growth of ovarian cancer cells [93], and treat respiratory infections [94]. Two BGCs in strains CchA and CchB each (cluster of *CchA_009358*, *CchA_013562*, *CchB_009236*, *CchB_014124*) were predicted for being responsible for the production of armillyl orsellinate 8α-hydroxy-6-protoilludene. This compound represents a crucial precursor for melleolides, a class of compounds with potential applications as fungicides, antimicrobials, and cancer therapeutics [95,96]. Most of the core genes of these above clusters were expressed according to the transcript data. It was indicated that *A. chaxingu* was able to produce plenty of terpenes. Many terpenoid compounds have been identified in the *C. cylindracea* species complex. A total of 22 volatile sesquiterpenes [91] were identified in *C. aegerita* at different growth stages, including δ-cadinene, cadinane-type sesquiterpenoids, and Δ6-protoilludene. These volatile sesquiterpenes were the precursors of numerous important active substances [97,98]. In *C. salicacola*, over 20 illudin-type sesquiterpenes [99,100,101,102], three aromadendrane-type sesquiterpenes [103], two Fomannosane-type sesquiterpenoids [101], and one bis-sesquiterpene [104] were isolated and identified. These compounds exhibited various antibacterial, antiviral, or antitumor activities [105]. Three protoilludene sesquiterpenes were isolated from *C. cylindracea* and exhibited significant inhibitory activity against pathogenic bacteria [106,107].

At present, polysaccharides [4] and steroid compounds [108] have been reported to be derived from *A. chaxingu*. Notably, six out of the eight steroid compounds identified from the genus *Agrocybe* were derived from the fruiting bodies of *A. chaxingu* [91]. The discovery of rich biosynthetic gene clusters for terpenes in *A. chaxingu* highlighted the necessity for further investigation into the structure and bioactivities of these compounds. Additionally, the complete genome of *A. chaxingu* will provide a crucial molecular foundation for the precise development of natural products and the analysis of metabolic pathways.

### 3.5. Whole Genome Comparison and Syntenic Analysis

A synteny analysis of the CchA and CchB genomes using MUMmer revealed that these two genomes exhibited a high overall level of synteny, although some minor regions displayed non-synteny (Figure 6A). Further analysis demonstrated that approximately 70.53% of the genes (E value < 1 × 10^−5^) on CchA exhibited synteny on the homologous chromosomes of CchB (Figure 6B). The highest gene collinearity of 83.09% was observed between chr 4 of the CchA genome and the homologous chromosomes of the CchB genome, while the lowest collinearity of 55.7% occurred on chr 9. Additionally, 858 genes in CchA exhibited collinearity with genes on non-homologous chromosomes in CchB (indicated by red in Figure 6B). It was indicated that the majority of these genes remained relatively stable during evolution, with only a few genes potentially having undergone gene rearrangements. The rearranged genes were found to be significantly enriched in a number of biological processes, including DNA synthesis, cell cycle regulation, antioxidant defense, and cellular protection (Appendix A). The rearrangement of genes has been a significant factor in the evolutionary process of individual organisms. This can be observed in the generation of new genes [109], the activation of genes [110], and the provision of valuable resources for phylogenetic studies [111].

Furthermore, the structural variations between homologous chromosomes of the two genomes were analyzed by SyRI. Structural variations of tens of kilobases in size were identified, including inversions, translocations, and duplications (Figure 6C). The collinear nucleotide sequences between the homologous chromosomes of the CchA and CchB genomes were found to be approximately 33.95 Mb in length, which was consistent with the collinearity ratio of genes. Compared to the CchB genome, the CchA genome exhibited 19 translocations, amounting to 519,232 bp, which accounted for 1.03% of the entire CchA genome. Additionally, there were 36 inversions spanning 1,655,412 bp, constituting 3.27%, as well as 15 duplications totaling 208,799 bp, which accounted for 0.41%. In the genomes of CchA and CchB, the observed rearranged genes and structural variation regions were rich in retrotransposons (Figure 2), so we speculated that these may have been caused by retrotransposons [46,112].

Through comparing the CchA and CchB genes, and considering completely missing genes as unique, the results revealed that there were 613 unique genes for CchA (4.26% of all the genes) and 483 for CchB (3.40% of all the genes). All these unique genes were distributed on each chromosome of both CchA and CchB (Figure 2). The majority of unique genes were classified as hypothetical proteins, representing 88.74% and 82.19% of the unique genes in CchA and CchB, respectively. In contrast, the proportion of hypothetical proteins within the overall genome was relatively low, comprising 17.42% for CchA and 16.61% for CchB. The KEGG enrichment analysis revealed that the unique genes of CchA were significantly enriched in DNA replication and RNA degradation (Appendix A), with a total of 32 genes involved in this process. The absence of substantial enrichment in other enrichment analyses may have been attributable to the high proportion of hypothetical proteins present in the unique genes of both CchA and CchB.

Among the unique genes in CchA with annotated functions, seven were annotated as leucine-rich repeat domain superfamily, 11 related to intracellular transport and metabolism, primarily encompassing carbohydrates, lipids, peptides, amino acids, nucleotides, and secondary metabolites. Two unique genes (*CchA_005693* and *CchA_005699*) encoding clitocypin, a fungal cysteine protease inhibitor, were identified in the genome of CchA. The PCR amplification (Appendix A) and sequencing have confirmed that these two genes were genome CchA specific. The function of these two genes is our next project. Among the annotated unique genes in CchB, a total of 36 genes were identified as being primarily involved in maintaining genome stability and the stress response. These included 15 genes involved in DNA replication, recombination and repair, nine genes associated with post-translational modification, five distinct types of zinc finger domains, and seven catalytic proteinase families. Moreover, among the unique genes associated with environmental adaptation and defense, there were 13 genes related to substance transport, nine genes involved in forming defensive cellular scaffolds, and eight genes associated with signal transduction.

Comparative analysis of haploid genomes within the same individual was performed in some species of edible fungi, including *L. edodes*, *Taiwanofungus camphoratus,* and *Tremella fuciformis*. In the compatible monokaryons SP3 and SP30 of *L. Edodes* [77], approximately 30% of their genomes were non-syntenic, with large-scale segmental rearrangements; their genomes contained 333 (2.91%) and 366 (3.25%) unique genes, respectively, which was lower than in this study. A substantial number of inversions, translocations, and duplications were identified between the genomes of monokaryons Wpm-1 and Wpm-4 of *L. Edodes* [113]. In *T. camphoratus* [114], the 2nd and 10th chromosomes of the monokaryon W1 underwent unequal and reciprocal rearrangements in the monokaryon V5, resulting in a longer 2nd chromosome and a shorter 10th chromosome in V5. In the compatible monokaryons A and B of *T. fuciformis* [115], rearrangements and sequence variations were observed in chromosomes 1, 2, and 5, resulting in the formation of distinct genome structures. The A and B genomes exhibited the presence of 100 (1.18%) and 134 (1.56%) unique genes, respectively. Obviously, high levels of rearrangements were typically observed between chromosomes of monokaryons in edible fungi, and these rearranged gene regions were often rich in retrotransposons [46]. Furthermore, each monokaryon contained its own unique genes, though the proportion of unique genes varied.

### 3.6. Phylogenomic Analysis and Gene Family Evolution

A phylogenetic analysis was conducted using the genomes of CchA and CchB, along with those of 13 other species from the Agaricales (Appendix A). Protein clustering in 15 fungi yielded 1735 single-copy orthologous ones. Maximum likelihood (ML) phylogenetic analysis based on these shared single-copy orthologous proteins produced a super matrix comprising 556,265 amino acid sites. Phylogenomic analysis supported that *Agrocybe* was not a monophyletic group, which was consistent with a previous study [116]. The two monokaryotic strains of *A. chaxingu* AS-5 clustered together with *A. chaxingu* strain GCA_027627235.1 (Figure 7). These three strains formed a group with *C. aegerita* GCA_902728275.1, and they all belong to the *C. cylindracea* species complex. Phylogenomic analysis supported *A. chaxingu* and *C. aegerita* as distinct species, although they were frequently confused in the market. Additionally, *A. pediades*, which is also classified within the genus *Agrocybe* along with *A. chaxingu*, clustered into a different clade. This result was consistent with the phylogenetic studies using different molecular markers [10,117,118,119].

The molecular clock method was used to estimate divergence times, with fossil records of *C. cinerea* and *L. bicolor* serving as calibration points [62]. The results revealed that the divergence time between *C. cylindracea* species complex and other species were estimated to be approximately 6.9–12.6 million years ago (MYA).

Since 2014 when the genus *Agrocybe* was changed to *Cyclocybe* [119], the strains in the *C. cylindracea* species complex have demonstrated diversity in their genus names, resulting in confusion between the use of *Agrocybe* and *Cyclocybe*. *A. aegerita* [11], *C. aegerita* [10], *C. cylindracea* [120], and *A. cylindracea* [12] were all in use. Moreover, the outcomes were suboptimal when straightforward molecular markers were employed to categorize strains within the *C. cylindracea* species complex. For example, the use of rDNA-ITS to classify species within the genus *Agrocybe* [117,118] revealed that species such as *C. salicacola*, *C. cylindracea*, *A. chaxingu*, and *C. aegerita* could not be clearly distinguished. Nevertheless, it was evident that there were discernible genetic variations between the *C. cylindracea* and *A. chaxingu* strains, as evidenced by the utilization of three molecular markers: the nuclear ribosomal unit, the mitochondrial SSU-rDNA, and the mitochondrial cob gene [121]. In addition, the use of ITS, the ribosomal RNA large subunit (LSU), the translation elongation factor 1-α gene (TEF1α), and the RNA polymerase II subunit gene (RPB2) as four molecular markers assisted in delineating the European species *C. aegerita* from a new complex of Asian taxa [10]. The development of multiple molecular markers based on whole-genome sequencing and comparative genomics analysis will facilitate the identification of genetic differences within this species complex [10].

### 3.7. Gene Expression Analysis

To investigate the gene expression profiles of different nuclei at the stages of mycelia (MY) and fruiting bodies (FB) of the strain *A. chaxingu* AS-5, RNA-seq was conducted, and a total of 38.5 Gb sequences were generated (Appendix A). Using the assembled CchA and CchB genomes as reference genomes, the RNA-seq data was analyzed (Figure 8). At both MY and FB stages, there was no significant difference between the average transcription levels of all genes in CchA and CchB (*p* > 0.05). 

Furthermore, the expression levels of unique genes in the genomes of CchA and CchB were focused. It was found that the expression levels of all genes at both MY and FB stages were significantly higher than those of unique genes associated with either CchA or CchB (*p* < 0.05). At both of these stages, the overall expression level of CchB unique genes was significantly higher than that of CchA unique genes (*p* < 0.05, Figure 8A).

At the MY and FB stages, 66.23% and 65.42%, respectively, of unique genes of CchA were not expressed (TPM < 1 for two of three repetitions), and only 42.65% and 42.44%, respectively, for CchB. Among the 613 unique genes in CchA, 382 of them (62.32%) were not expressed at both stages. These genes may be environmental response genes or genes that are regulated during specific developmental stages, such as spore germination, dormancy, or other developmental stages. A total of 24 unique genes of CchA were expressed exclusively at the FB stage (Figure 8B), 23 of these genes were identified as hypothetical proteins, and one was involved with ribonuclease H-like protein. A total of 19 unique genes were expressed only at the MY stage and not at the FB stage, which may have been related to nutrient absorption, growth, and environmental adaptation of the mycelia (Figure 8C). A total of nine were expressed at high levels (TPM > 100 for two of three repetitions, Figure 8D) at both stages, including *CchA_008179*, *CchA_001116*, *CchA_001413*, *CchA_002197*, *CchA_011947*, *CchA_008656*, *CchA_000041*, *CchA_010495*, and *CchA_012039*. These genes are likely to be associated with fundamental metabolic processes, the synthesis of structural proteins, and other related functions. 

Similarly, among the 483 unique genes in CchB, 191 genes (39.54%) were not expressed and 29 genes were expressed at high levels (Figure 8G) at both stages. Additionally, 14 unique genes were expressed only at the MY stage (Figure 8F), and 15 unique genes were expressed only at the FB stage (Figure 8E). The 15 genes associated with fruiting body formation were all hypothetical proteins. Among them, five were unique hypothetical proteins found only in the *C. cylindracea* species complex, such as *CchB_007110*, *CchB_003173*, *CchB_011365*, *CchB_000013*, and *CchB_003333*.

A total of 39 unique genes in CchA and CchB were exclusively expressed at the FB stage (Appendix A, Figure 8B,E). The identification of these genes could provide a valuable genetic resource for the study of the fruiting body development mechanism of *A. chaxingu*.

Consequently, compared to CchA, we found that at both the MY and FB stages, CchB unique genes exhibited higher expression levels and a lower proportion of genes that were not expressed at both stages. This indicated that CchB was likely to have played a more significant role than CchA during the mycelium growth and fruiting body development. Finally, it was demonstrated that CchA and CchB expressed unique genes exclusively during the FB stage, indicating that CchA and CchB required coordinated regulation in the fruiting body development of *A. chaxingu*.

## 4. Discussion

This study assembled and compared the T2T genomes of a pair of sexually compatible monokaryons, CchA and CchB, to understand the genetic characteristics of *A. chaxingu*. Gene transcription at the stages of mycelium and fruiting body were also compared to investigate the genes involved in fruiting body development. The CchA and CchB T2T genomes represented the only high-quality genomes of *A. chaxingu* available until now. The T2T genome facilitated more accurate identification of structural variations and single nucleotide variations, particularly complex structural variations and variations in repetitive sequences. This was of significant importance for studying genetic diversity, evolutionary mechanisms, and adaptability.

It was found that 70.53% of the genes were collinear between homologous chromosomes in the two genomes, and 858 genes from CchA were rearranged to non-homologous chromosomes in CchB. A comparison of the two monokaryons showed that CchA contained 4.26% unique genes, while CchB had 3.4%. The genetic characteristics of these two sexually compatible monokaryon genomes revealed both similarities and differences when compared to other edible and medicinal fungi. For instance, the collinear gene ratio was similar to that observed in *L. edodes* [77], but the proportion of unique genes was higher than that identified in *L. edodes* (2.91% and 3.25%) [77] and *T. fuciformis* (1.18% and 1.56%) [115]. Furthermore, chromosomal rearrangement phenomena were observed in *A. chaxingu*, *L. edodes* [77,113], *T. fuciformis* [115], and *T. camphoratus* [114]. This suggested that the two sexually compatible nuclei in dikaryotic edible fungi exhibit distinctive genetic characteristics, primarily reflected in disparate genetic information and chromosomal rearrangements.

The formation of dikaryons is subject to the regulation of mating-type genes, and *A. chaxingu* displayed distinctive characteristics in its mating-type genes. A difference in the *CchA_000467* gene at the *MatA* locus was identified between the two monokaryons, and the occurrence of *CchA_000467* being adjacent to the *Hd2* gene was exclusive to the *C. cylindracea* species complex. Furthermore, the *β-fg* gene in the *A. chaxingu* genome was not linked to the *MatA* locus, contrasting with many other edible fungi.

A comparative analysis of gene expression during the mycelium and fruiting body stages of *A. chaxingu* revealed that overall gene expression levels were not significantly different between the two monokaryons. However, the overall expression level and proportion of unique genes in CchB were significantly higher than those in CchA. Moreover, CchA and CchB exhibited distinct gene expression patterns during the fruiting body stage. This indicated that *A. chaxingu* required the coordinated control of both nuclei during fruiting body development, with CchB likely playing a more significant role. A similar phenomenon was observed during the fruiting body development of *L. edodes* [113], where the two nuclear types coordinated fruiting body development through the dominant expression of different functional genes. Currently, transcriptomics enables the study of dikaryotic edible fungi at various growth and development stages. However, research treating the two nuclei in dikaryons as distinct subjects remains limited [122] In *A. bisporus*, single-nucleotide polymorphism markers were employed to investigate the expression levels of allelic pairs between the two nuclei of dikaryons, elucidating the dominance-recessiveness relationships and complementary effects between the dikaryotic nuclei [123].

Several scientific issues in the edible mushroom industry require further consideration, such as strain degeneration and maintenance of strain characteristics, as well as the theoretical basis for hybrid breeding. These issues were closely related to the regulation and interaction of the two different types of nuclei in dikaryotic mushroom cells. Although dikaryotic gene expression often mirrors the dominance observed in one of the monokaryons [123], the division of labor between nuclei in dikaryons is crucial. Furthermore, the other nucleus plays a vital role in maintaining the stability of the dikaryon [124] Our findings revealed the unique genetic characteristics and expression profiles between the two sexually compatible monokaryon genomes of *A. chaxingu*, which is of significant importance for the genetic research.

Therefore, the two monokaryons of *A. chaxingu* each possess their own independent regulatory systems during dikaryon development, while also allowing for potential mutual regulatory information exchange and additive effects in expression levels. This study has highlighted the distinctive genetic characteristics of the two monokaryons of *A. chaxingu* and provided insights into fruiting body development.

## 5. Conclusions

In this study, we employed multiple techniques to sequence the complete genomes of two sexually compatible monokaryons, CchA and CchB, derived from a widely cultivated strain of *A. chaxingu* AS-5. For the first time, we successfully assembled two gap-free T2T genomes, and analyzed the genetic differences between them, along with their expression levels in growth and development. These high-quality genome assemblies serve as valuable resources for investigating the mechanisms underlying fruiting body development in *A. chaxingu*. They also open up new avenues for molecular breeding, genetic regulation of important agronomic traits, localization of quantitative trait loci for key genes, and the development of linked molecular markers for specific traits in *A. chaxingu*.

## Figures and Tables

**Figure 1 jof-10-00602-f001:**
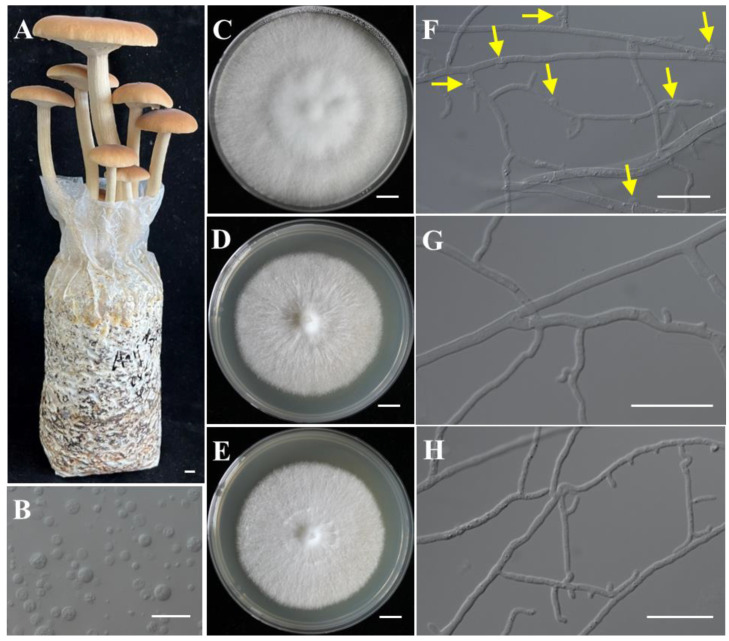
The fruiting bodies of *Agrocybe chaxingu* and the monokaryotic strains used for genome sequencing. (**A**) The fruiting bodies of *A. chaxingu*. (**B**) The protoplasts of *A. chaxingu*. (**C**–**E**) Colony of heterokaryotic and two monokaryotic strains. (**F**) Hyphae of heterokaryotic strain with clamp connections. Yellow arrows indicate the clamp connections. (**G**,**H**) Hyphae of monokaryotic strain CchA and CchB. Bars: (**A**,**C**–**E**) = 1 cm; (**B**,**F**–**H**) = 20 μm.

**Figure 2 jof-10-00602-f002:**
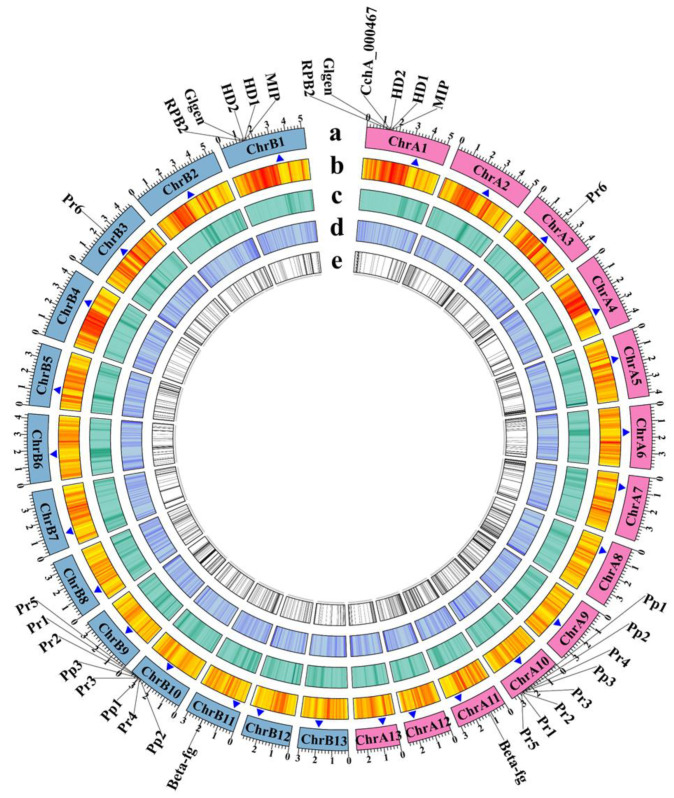
Genome features of *Agrocybe chaxingu* strains CchA and CchB. The genes labeled outside the circle were related to mating types and their adjacent genes. HD: Homeodomain, MIP: mitochondrial intermediate peptidase, Glgen: glycosyltransferase family 8 protein, RPB2: RNA polymerase II the second large subunit, Beta-fg: β-flanking, Pr: pheromone receptor, Pp: pheromone precursor. (**a**) The chromosomes of strains CchA and CchB (Mb). Pink represents the chromosomes of CchA. Light blue represents the chromosomes of CchB. (**b**) Gene density. (**c**) Repetitive sequence density. (**d**) GC content. (**e**) Unique genes. The blue triangles represent the possible centromere regions.

**Figure 3 jof-10-00602-f003:**
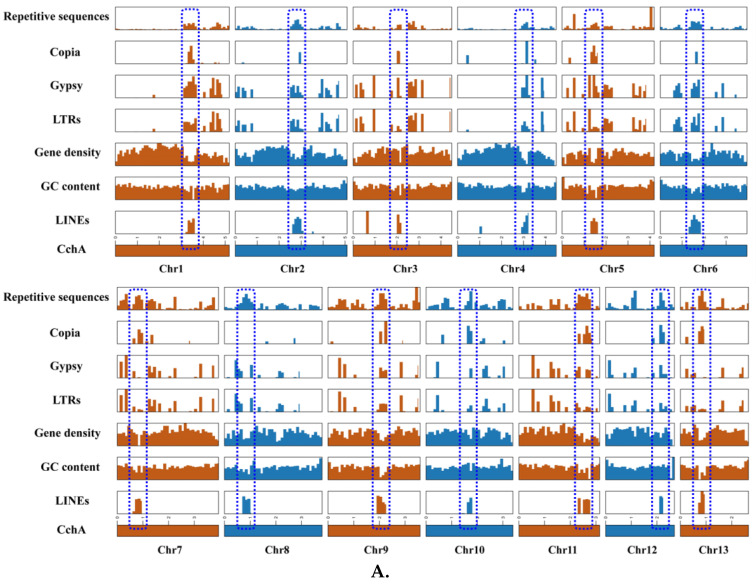
Features of the major repetitive sequences, GC content (within a 10 kb window), and gene density (within a 100 kb window) in the *Agrocybe chaxingu* CchA (**A**) and CchB (**B**) genomes. The blue dashed box represents the predicted centromere region.

**Figure 4 jof-10-00602-f004:**
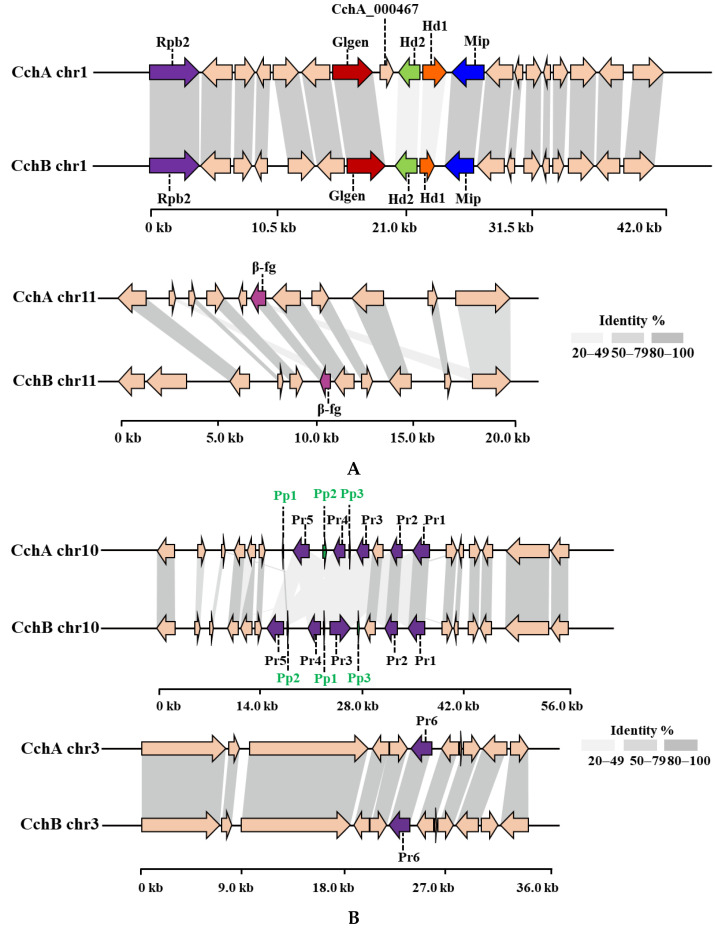
The protein synteny around the *MatA* (**A**) and *MatB* (**B**) loci of CchA and CchB. (**A**) Hd: Homeodomain, Mip: mitochondrial intermediate peptidase, Glgen: glycosyltransferase family 8 protein, Rpb2: RNA polymerase II, the second large subunit, β-fg: β-flanking. (**B**) Pr: pheromone receptor, Pp: pheromone precursor.

**Figure 5 jof-10-00602-f005:**
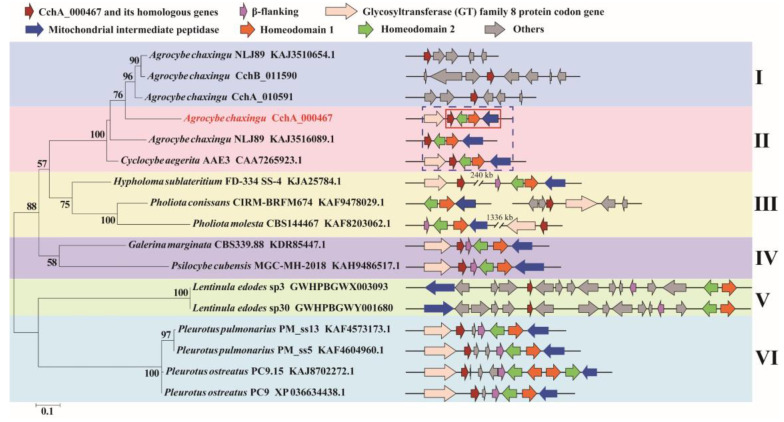
Phylogenetic analysis based on the amino acid sequences of the CchA_000467 and its homologs from 15 strains of basidiomycetes. The phylogenetic tree was constructed using RAxML (version 8) with 1000 bootstrap replicates. The numbers at the nodes represent the bootstrap percentages. The red fonts indicate the CchA_000467 proteins. The red solid box indicates that the *CchA_000467* gene was adjacent to the mating-type A locus. The blue dashed box indicates that in the *Cyclocybe cylindracea* species complex, the *CchA_000467* gene and its homologs were adjacent to the mating-type A locus. The letters Ι, ΙΙ, ΙΙΙ, ΙV, V, and VΙ represent several different types of positional relationships between the *CchA_000467* gene and its homologs with the mating-type A locus. In the phylogenetic tree, the text represents species names, strain identifiers, and GenBank numbers from the National Center for Biotechnology Information, excluding strains of CchA and CchB.

**Figure 6 jof-10-00602-f006:**
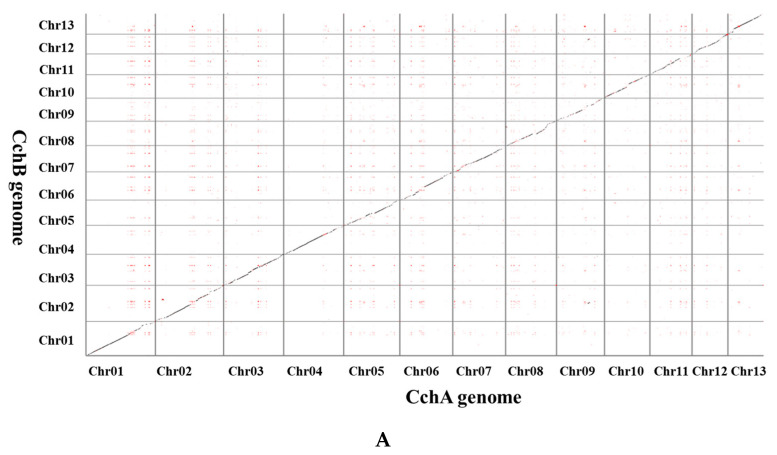
Whole-genome comparison between the assemblies of CchA and CchB. (**A**) Dot plot for the syntenic blocks. The black dots represent syntenic regions. The red dots represent the repetitive sequences in the genomes. (**B**) The gene collinearity between each chromosome of CchA and the 13 chromosomes of CchB. The blue lines represent the gene collinearity between homologous chromosomes. The red lines represent the gene collinearity between non-homologous chromosomes. (**C**) Chromosome-level local sequence differences. The triangle represents the telomere.

**Figure 7 jof-10-00602-f007:**
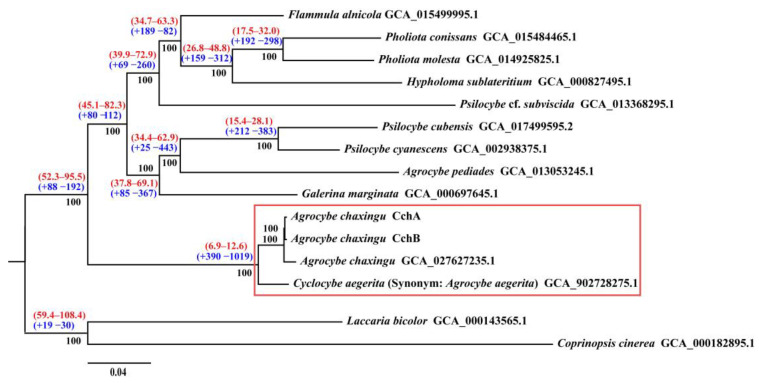
Phylogenomic analysis among 15 fungal genomes. Bootstrap values are indicated in black, divergence time with 95% CI (confidence interval) in red, and gene family expansion/contraction in blue. The red boxes represent the *Cyclocybe cylindracea* species complex.

**Figure 8 jof-10-00602-f008:**
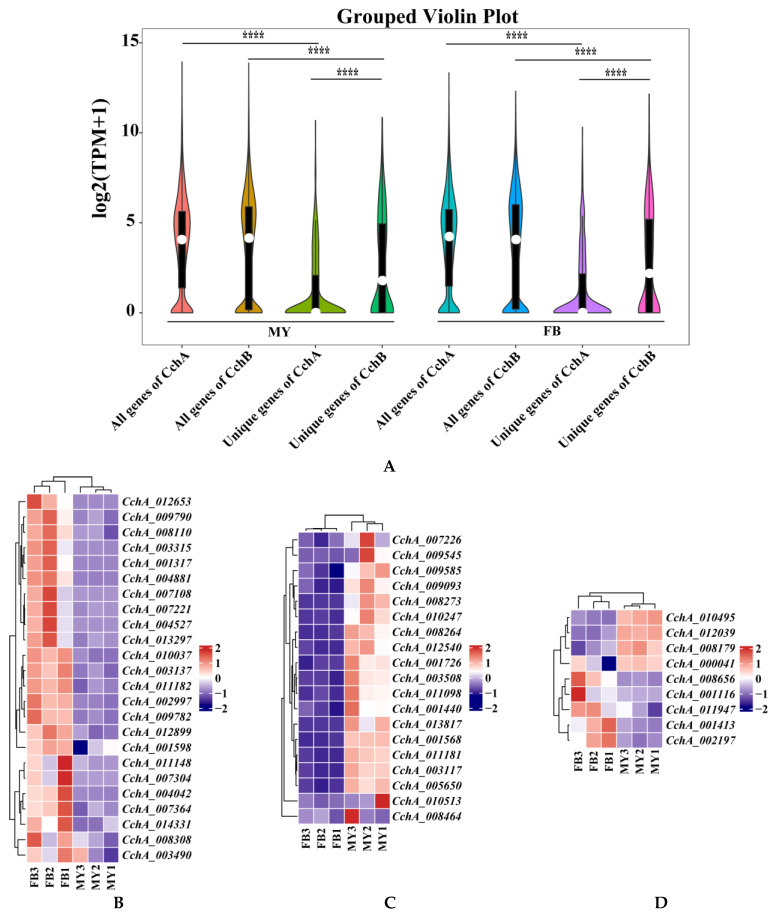
Transcriptome analyses of the strain *Agrocybe chaxingu* AS-5 between the mycelia and fruiting bodies. (**A**) Violin plot showing TPM values of the CchA and CchB in MY and FB. MY: mycelia; FB: fruiting bodies. The significance of any differences was assessed by *T* test. **** represented a significant difference, *p* < 0.05. (**B**) Expression of 24 CchA unique genes exclusively at the FB stage. (**C**) Expression of 19 CchA unique genes exclusively at the MY stage. (**D**) Nine CchA unique genes expressed at high levels at both stages of MY and FB. (**E**) Expression of 15 CchB unique genes exclusively at the FB stage. (**F**) Expression of 14 CchB unique genes exclusively at the MY stage. (**G**) Twenty-nine CchB unique genes expressed at high levels at both stages of MY and FB.

**Table 1 jof-10-00602-t001:** BUSCO assessment results in the genomes of CchA and CchB.

Type	Percent (%) of CchA	Percent (%) of CchB
Complete BUSCOs (C)	98.50	98.50
Complete and single-copy BUSCOs (S)	97.80	97.80
Complete and duplicated BUSCOs (D)	0.70	0.70
Fragmented BUSCOs (F)	0.10	0.10
Missing BUSCOs (M)	1.40	1.40

**Table 2 jof-10-00602-t002:** Global statistics for the genomes of CchA, CchB and the previously published strains of *C. cylindracea* species complex.

Assembly Feature	*Agrocybe chaxingu* CchA	*Agrocybe chaxingu* CchB	*Agrocybe chaxingu* MP-N11	*Cyclocybe aegerita* AAE-3	*Cyclocybe aegerita* AAE-3-13	*Cyclocybe aegerita* AAE-3-32	*Cgrocybe cylindracea* ASM1337643v1
Number of contigs	13	13	4466	122	120	120	3790
Assembly length (Mb)	50.60	51.66	40.30	44.79	44.74	44.73	56.49
Total N counts	0	0	31,056	1,418,454	975,332	1,109,304	940,733
Contig N50 (Mb)	3.95	3.97	0.0181	0.7684	0.7683	0.7683	0.5473
Max length (Mb)	5.19	5.46	0.12	2.76	2.76	2.76	2.74
Min length (Mb)	2.65	2.98	0.0005	0.0024	0.0033	0.0033	0.0005
Mean length (Mb)	3.89	3.97	0.0090	0.37	0.37	0.37	1.49
Complete BUSCOs (%)	98.50	98.50	/	97	97	94	/
Repeat content (%)	16.97	17.97	/	/	/	/	12.54
GC (%)	50.99	50.98	51.00	49.22	49.66	49.42	50.20
Assembly level	Chromosome-level	Chromosome-level	Scaffold	Scaffold	Scaffold	Scaffold	Scaffold
Number of telomeres	26	26	0	0	0	0	0
Reference	This study	This study	NCBI *: GCA_027627235.1	[11]	[11]	[11]	[12]

*: NCBI stands for the National Center for Biotechnology Information.

**Table 3 jof-10-00602-t003:** Classification of the repeat sequences in the genome of CchA and CchB.

Classification	Order	Super Family	CchA			CchB		
			Number of Elements	Length Occupied (bp)	Percentage of Sequence (%)	Number of Elements	Length Occupied (bp)	Percentage of Sequence (%)
Class I (Retroelements)			1125	2,633,630	5.21	1118	3,122,510	6.04
	SINEs		40	13,193	0.03	45	13,789	0.03
	LINEs		494	874,092	1.73	457	858,312	1.66
		RTE/Bov-B	0	0	0.00	41	10,852	0.02
		L1/CIN4	8	3451	0.01	0	0	0.00
		Unknown	486	870,641	1.72	416	847,460	1.64
	LTR elements		591	1,746,345	3.45	616	2,250,409	4.36
		Ty1/Copia	200	214,240	0.42	133	212,896	0.41
		Gypsy/DIRS1	391	1,532,105	3.03	483	2,037,513	3.94
Class II (DNA transposons)			130	156,143	0.31	174	300,689	0.58
	Hobo-Activator		0	0	0.00	95	149,817	0.29
	Tc1-IS630-Pogo		14	3212	0.01	0	0	0.00
	Unknown		116	152,931	0.30	79	150,872	0.29
Unclassified			6141	4,967,145	9.82	6886	5,258,353	10.18
Total interspersed repeats			7396	7,756,918	15.33	8187	8,681,552	16.81
Rolling-circles			205	514,795	1.02	128	275,812	0.53
Satellites			1	87	0.00	0	0	0.00
Simple repeats			5842	257,760	0.51	5979	262,399	0.51
Low complexity			1073	59,876	0.12	1073	61,607	0.12
Total repeats			14,517	8,589,436	16.97	15,367	9,281,370	17.97

## Data Availability

The raw sequences of PacBio Hifi were submitted to NCBI SRA (http://www.ncbi.nlm.nih.gov/sra, accessed on 31 July 2024) under BioProject accession numbers PRJNA1142205.

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
