# Peer review of "Telomere-to-Telomere Haplotype-Resolved Genomes of Agrocybe chaxingu Reveals Unique Genetic Features and Developmental Insights"

_jof, 2024, doi:10.3390/jof10090602_

Round 1

Reviewer 1 Report

The manuscript by Xutao Chen et al.  ” Telomere-to-telomere Haplotype-resolved Genomes of Agrocybe chaxingu Reveals Unique Genetic Features and Developmental Insights” give a comprehensive picture about the structure of the genome of two monokaryotic Agrocybe chaxingu strains and the expression of genes in the dikaryon formed by the monokaryons.

Just a few small comments. As an unclear point remains what the authors think is the B-mating-type structure, which of the Pr genes belong to the mating type which are outside the mating type ( see Wirth et al. J. Fungi (Basel). 2021 May 20;7(5):399. doi: 10.3390/jof7050399 ). I also wonder, whether “In S. commune [45] and C. cinerea [79,80], multiple Hd domains were observed in regions apart from the MatA” is a right comment.

It is of great interest to see how much difference in the number of unique genes and in arrangements between two compatible mates can be. Is it possible to call them any more homokaryons? Although, the expression of the unique genes of the monokaryons during the mating were carefully analyzed, a question remained whether there are also differences in the expression between alleles of the genes expressed in both monokaryons (see Gehrmann et al. Proc Natl Acad Sci U S A. 2018 doi: 10.1073/pnas.1721381115).

No detailed comments

Reviewer 2 Report

The submitted manuscript is devoted to the genome sequencing of two sexually compatible monokaryons A. chaxingu – CchA and CchB – that were derived from a widely cultivated strain A. chaxingu AS-5. The study performed by the authors is well designed, properly described and comprehensively illustrated. The obtained genomes are of the outstanding telomere-to-telomere quality. Moreover, the authors performed an additional transcriptomic study at the mycelial and fruiting body stages; this study makes the scope of the submitted article far exceed the format of a genome report alone.

Overall, this manuscript can be published as is, without further revisions. However, I would suggest adding extra information in lines 100–101, since the only available description of the “protoplast monokaryotyzation” procedure is written in Chinese. I strongly recommend to comprehensively describe the “protoplast monokaryotyzation” procedure in the current manuscript; this will drastically improve the reproducibility of the authors' results, and reproducibility is extremely important in any scientific study.

The grammar and style of the English are generally of good quality; only minor corrections of some typos are needed.

Again, this is an excellent article that I read with pleasure and joy. It should definitely be published.

lines 100–101: since the only available description of the “protoplast monokaryotyzation” procedure is written in Chinese, I strongly recommend to comprehensively describe the “protoplast monokaryotyzation” procedure in the current manuscript.
